

# Ice fog observed at cirrus temperatures at Dome C, Antarctic Plateau

Étienne Vignon[1], Lea Raillard[1], Christophe Genthon[1], Massimo Del Guasta[2], Andrew J. Heymsfield[3], Jean-Baptiste Madeleine[1], and Alexis Berne[4]

[1]Laboratoire de Météorologie Dynamique/IPSL/Sorbonne Université/CNRS, UMR 8539, Paris, France
[2]INO-CNR, Sesto Fiorentino, Italy
[3]National Center for Atmospheric Research, Boulder, Colorado
[4]Environmental Remote Sensing Laboratory (LTE), École Polytechnique Fédérale de Lausanne, Switzerland

**Correspondence:** É. Vignon (etienne.vignon@lmd.ipsl.fr)

**Abstract.**

As the near-surface atmosphere over the Antarctic Plateau is cold and pristine, its physico-chemical conditions resemble to a certain extent those of the high-troposphere where cirrus clouds form. In this paper, we carry out an observational analysis of two shallow fog clouds forming in-situ at cirrus-temperatures - that is, temperatures lower than 235 K - at Dome C, inner Antarctic Plateau. The combination of lidar profiles with temperature and humidity measurements from advanced thermo-hygrometers along a 45-m mast makes it possible to characterise the formation and development of the fog. High supersaturations with respect to ice are observed before the initiation of fog and the values attained suggest that the nucleation process at play is the homogeneous freezing of solution aerosol droplets. To our knowledge, this is the first time that in situ observations show that this nucleation pathway can be at the origin of an ice fog. Once nucleation occurs, the relative humidity gradually decreases down to subsaturated values with respect to ice in a few hours, likely owing to vapour deposition onto ice crystals and turbulent mixing. The development of fog is tightly coupled with the dynamics of the boundary-layer which, in the first study case, experiences a weak diurnal cycle while in the second case, transits from a very stable to a weakly stable dynamical regime. Overall, this paper highlights the potential of the site of Dome C for carrying out observational studies of cloud microphysical processes in natural conditions and using in-situ ground-based instruments.

## 1 Introduction

Given the prevailing very cold temperatures and the pristine air, the near-surface atmosphere over the Antarctic Plateau shares similarities with the high-troposphere in terms of physico-chemical conditions. Antarctic ice fogs, i.e. near-surface ice clouds, have been observed at typical cirrus temperatures namely T < 235 K (Ricaud et al., 2017), and it is therefore legitimate to question to what extent their microphysical properties resemble those of cirrus clouds. Ice fogs have non-negligible effects on polar climates as their radiative forcing has significant impacts on the surface energy budget (e.g. Blanchet and Girard, 1995). Ice fogs can be distinguished from another surface ice cloud type called 'diamond dust' through visibility criteria - fogs are optically deeper - or ice crystal properties such as their size and their number concentration. As stressed in Gultepe





et al. (2017), this classification is subjective and the distinction between ice fogs and diamond dust is somewhat blurred. In the present paper, we use the term 'ice fogs' independently for fog and diamond dust.


Most of polar ice fogs form through the advection of warm and moist air from the mid-latitudes followed by radiative cooling (e.g. Curry et al., 1990). The formation of fog is also generally pre-conditionned and accompanied by a stable stratification in the boundary layer (Gultepe et al., 2015) which maintains the humidity and the low temperatures near the surface. Gultepe et al. (2017) summarise the microphysical and dynamical processes through which ice fogs develop. Ice fog crystals form by either

homogeneous or heterogeneous nucleation. In the former case, supercooled liquid droplets produced at higher temperatures through Cloud Condensation Nuclei (CCN) activation, - e.g. above the open ocean - are advected into a colder environment - e.g. above a continental surface, the sea-ice or an ice-sheet - and then freeze when $T < 235$ K. In the second case, supercooled liquid droplets can freeze at higher temperature if they contain or enter in contact with Ice Nuclei Particles (INPs). In addition to these so-called immersion and contact freezing processes, heterogeneous nucleation might occur without the presence of

supercooled liquid droplets through the direct deposition of water vapor onto INPs but the occurrence of such a process in the atmosphere is still debated (Marcolli, 2014). Given the low concentrations of INPs over the Antarctic (Belosi et al., 2014), supercooled liquid droplets can be observed at temperatures down to $248$ K (Silber et al., 2019; Ricaud et al., 2020) and they were shown to be at the origin of fogs over the Antarctic coast (Kikuchi, 1971, 1972).

Ice fogs forming locally over the Antarctic Plateau when the temperature remains well below 235 K have received much less attention hitherto. In this cirrus-temperature regime, no supercooled water droplets are present and the fog necessarily cannot have a liquid origin. In this particular case, ice crystals may form through freezing of solution aerosol particles also named haze droplets (Heymsfield and Sabin, 1989; Girard and Blanchet, 2001) when the relative humidity reaches a value that depends on the particle size and water activity. This value is above the saturation with respect to ice - so the air is supersaturated with

respect to ice - but is lower than the liquid water saturation (Koop et al., 2000; Baumgartner et al., 2022). There is also evidence that solution aerosol particles can also freeze heterogeneously due to the presence of INPs (DeMott et al., 1998; Kärcher and Lohmann, 2003) but this process remains poorly understood especially at very low temperatures (Heymsfield et al., 2017).

The air near the surface of the Antarctic Plateau frequently experiences high supersaturations with respect to ice (Genthon

et al., 2017). The evidence of such a phenomenon is quite recent as conventional capacitive thermo-hygrometers deployed on weather stations fail to report supersaturation because the excess of water vapour with respect to saturation condenses on the sampling device and the sensor (Genthon et al., 2017). The recent development and deployment of advanced thermo-hygrometers able to sample supersaturations on a 45-m mast (Genthon et al., 2022) at the French-Italian Concordia station on the Dome C, East Antarctic Plateau, paves the way for an examination of the humidity evolution during ice fog formation and

could give insights into the microphysical processes - including the nucleation and growth of ice crystals - that are potentially involved.



The objective of the present paper is to study the development of ice fog that do not have a liquid origin and do not correspond to maritime advections but that form locally at $T < 235$ K over the East Antarctic Plateau. Two case studies are analysed in details through an in depth examination of meteorological data at Dome C, site particularly known for its very stable boundary layers and extreme temperature inversions (Vignon et al., 2017).

## 2    Data and methods

The French-Italian Concordia station is located on the Dome C, high Antarctic Plateau (75°06' S, 123°20' E, 3233 m a.s.l, Local Time LT = UTC+8 h). The landscape is a homogeneous and flat snow desert where the monthly mean 2 m temperatures ranges from about 246 K in austral summer to about 208 K in the polar night in winter (Genthon et al., 2021). During austral summer, the atmospheric boundary layer experiences a marked diurnal cycle with an alternation of diurnal shallow convection - when the sun is high above the horizon - and nocturnal stable stratification. Conversely during winter, the boundary-layer is almost always stably - even very stably - stratified (Genthon et al., 2013). The absence of terrain slope precludes the local generation of katabatic winds. The near-surface wind is mostly south-southwesterly and the annual 3 m mean speed is 4.5 m s$^{-1}$ (Argentini et al., 2014). Occurrences of significant wind-transported snow events are seldom (Libois et al., 2014). The sets of observational data collected at the station and used in this study are described in the following sub-sections.

### 2.1    Wind, temperature and radiative flux measurements

Wind and temperature measurements are performed at six levels on a 45-m mast located 1-km upwind of Concordia station. Temperature is measured using mechanically-ventilated Vaisala HMP-155 thermo-hygrometers and wind speed and direction are obtained with R. M. Young 05103 aerovanes. 30-min average data have been used in this paper. Details on data acquisition and processing are given in (Genthon et al., 2021). The downward long-wave and short-wave radiative fluxes are measured at the Dome C Baseline Surface Radiation Network (BSRN) station using two Kipp and Zonen CM22 secondary standard pyranometers and two Kipp and Zonen CG4 pyrgeometers (Lanconelli et al., 2011; Driemel et al., 2018).

### 2.2    Humidity measurements

The near-surface atmosphere over the Antarctic Plateau is frequently supersaturated with respect to ice, which prevents from correctly measuring humidity with conventional capacitive hygrometers since the excess of moisture with respect to saturation condenses once the air gets in contact with the sampling device or the sensor (Genthon et al., 2017). The humidity data used in this study were obtained with advanced HMP-155 thermohygrometers deployed at three levels on the 45-m mast (Genthon et al., 2022). In a nutshell, the innovative technology behind these adapted HMP-155 consists of heating the air aspirated in the intake such that the sensor always samples an under-saturated air. A separate thermometer measures the ambient air temperature in order to retrieve the relative humidity with respect to liquid phase (RHl) of the ambient air. Relative humidity with respect to ice (RHi) is then calculated using the Murphy and Koop (2005)'s saturation vapor pressure formulae. It is worth noting that these innovative and low-cost instruments compare very satisfactorily with 'reference' frost point hygrometers but unlike the



latter, they are further capable to operate in extremely cold conditions, even at temperatures < 220 K, which frequently occur at Dome C. Estimations of RHl and RHi uncertainties associated with the measurements are provided in Appendix A. Further

details on the measurement system and the humidity data are provided in Genthon et al. (2017, 2022).

## 2.3   Radiosounding data

Vaisala RS92 radiosondes are released every day at 2000 LT (1200 UTC) from the Routine Meteorological Observation program (http://www.climantartide.it/) and processed with the standard Vaisala evaluation routines. Note that the actual launching time is around 1900 such that the sonde reaches the tropopause at 2000 LT. No temperature and humidity correction for time-

lag errors has been applied since our analysis mostly focuses on the near-surface atmosphere (where the time-lag effect is negligible, Tomasi et al., 2012). Moreover, the relative humidity bias for standard RS92 measurements is relatively small: lower than 5% in magnitude across the whole troposphere at Dome C according to Tomasi et al., 2012.

## 2.4   Lidar measurements

A tropospheric depolarization aerosol lidar (532 nm) has been operating at Dome C since 2008 (http://lidarmax.altervista.org/

englidar/Antarctic%20LIDAR.php). The lidar provides 5 min tropospheric profiles of aerosols and clouds continuously, from 20 to 7000 m a.g.l., with a 7.5 m vertical resolution. Further technical details are given in (Palchetti et al., 2015; Ricaud et al., 2020). Note that only the lidar backscattering signal is shown in this study but the investigation of the depolarisation ratio (not shown) reveals high values (>10%) for the two fog events analysed in this study. This suggests that the ice fogs do not contain supercooled liquid drops, which is consistent with the temperature (<235 K) at which they are observed. To support

the interpretation of lidar data, time-lapse webcam videos of local sky conditions are also collected.

## 2.5   Back-trajectory analysis

To ensure that the studied fog events correspond to local cloud formation over the Antarctic Plateau and are not associated with maritime air intrusions, we estimate air parcel Lagrangian back trajectories using the HYSPLIT modeling system (https://www.ready.noaa.gov) applied to the Global Data Assimilation System analysis of the National Center for Environmental

Prediction with a horizontal grid of $0.5^{\circ} \times 0.5^{\circ}$ and a hourly temporal resolution. We calculate 2-day trajectories starting - backward in time - at the 4 closest grid points to Dome C and at two different heights near the surface: 50 m a.g.l. and 100 m a.g.l. We will see hereafter that the maximum ice fog depth during the events of interest is about 200 m. Assuming a fall velocity of $\approx 1 \, \mathrm{cm} \, \mathrm{s}^{-1}$, ice crystals forming at the top of the fog layer reach the ground in less than 6 h. Therefore, a 2-day trajectory duration is sufficient to track the trajectory of all ice crystals observed above Dome C.




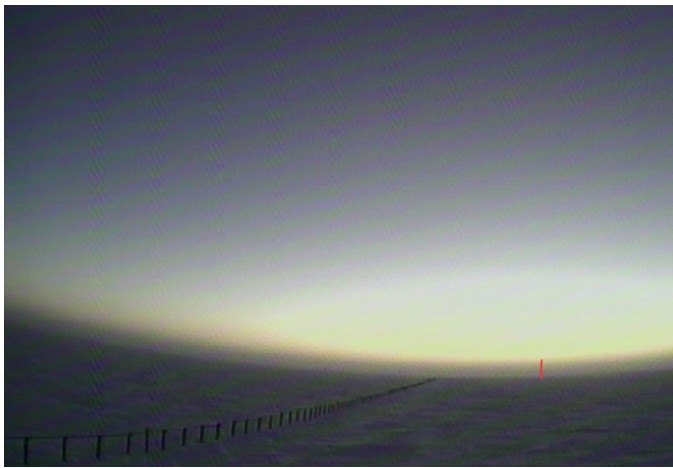

**Figure 1.** Webcam image of the ice fog at 12 UTC (20 LT), 8 March 2018 at Dome C. The shallow and thin fog layer manifests as a thin dark band above the horizon (delimited with a red line).

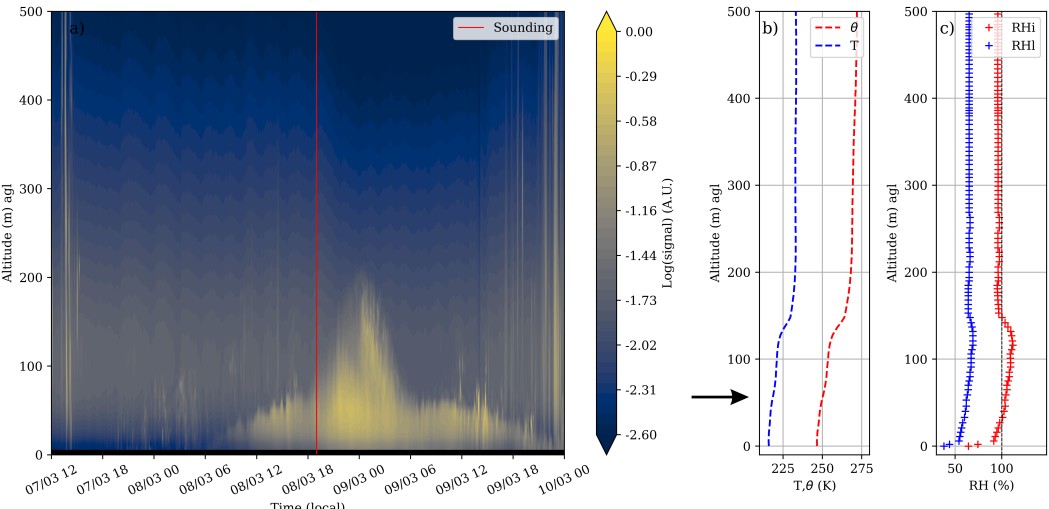

**Figure 2.** Panel a: Time-height plot of the lidar backscattering signal intensity. The red line indicates the radiosonde launching time. Panel b shows the vertical profile of temperature (blue) and potential temperature (calculated with a reference pressure of $10^5$ Pa, red) from the radiosounding. The black arrow indicates the top of the boundary layer. Panel c shows the vertical profile of relative humidity with respect to ice (RHi, red) and with respect to liquid water (RHl, blue) from the same radiosounding.



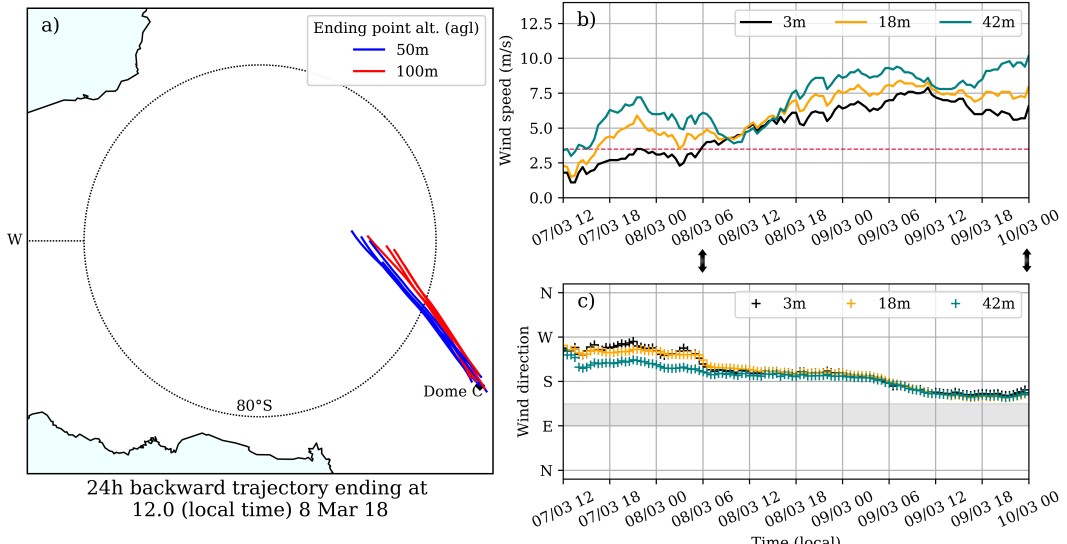

**Figure 3.** Panel a: 2-day back-trajectories ending at 50 m a.g.l. (blue) and 100 m a.g.l. (red) at the closest-to-Dome C four grid points. Panels b and c show the time series of the wind speed and origin at three levels on the mast. The dashed red line in panel b indicates the 3-m wind speed threshold value for snowdrift triggering (see details in the main text). The grey band in panel c delimits the wind direction range in which the flow can be possibly affected by station exhausts. Black arrows indicate the times at which the fog visually appears and disappears near the surface in the lidar measurements.

## 3 Results and discussion

### 3.1 Event 1: 7-9 March 2018

#### 3.1.1 Overview

The first ice fog event we focus on occurred between the 7 and 9 March 2018. A picture showing the landscape of Dome C in the shallow and thin fog during sunset, 8 March 2018 is shown in Fig. 1. Fig. 2 shows a time-height plot of the lidar backscattering signal during the event. A clear shallow fog layer starts to develop from about 0600 UTC, 8 March and grows from the ground until it reaches a maximum height of 200 m at 0100 LT, 9 March. Its depth then sharply decreases until 0600 LT and the decay continues until dissipation at 00 LT, 10 March.

The radiosounding reveals a layer supersaturated with respect to ice from 40 m to 150 m a.g.l. The maximum RHi value of 120 % is reached at 130 m a.g.l which roughly corresponds to the upper limit of the fog layer and of the boundary-layer top inversion at this time. The temperature between the ground and 120 m ranges between 216 K at the bottom and 230 K at the top (Fig. 2b) which confirms that the fog develops in cirrus-like temperature conditions.

The 3-m wind slightly veers from a south-westerly to a south-easterly direction and its magnitude increases throughout the event from 2 to 7.5 m s$^{-1}$. Libois et al. (2014) identify drifting snow events at Dome C when the 10-m wind speed exceeds

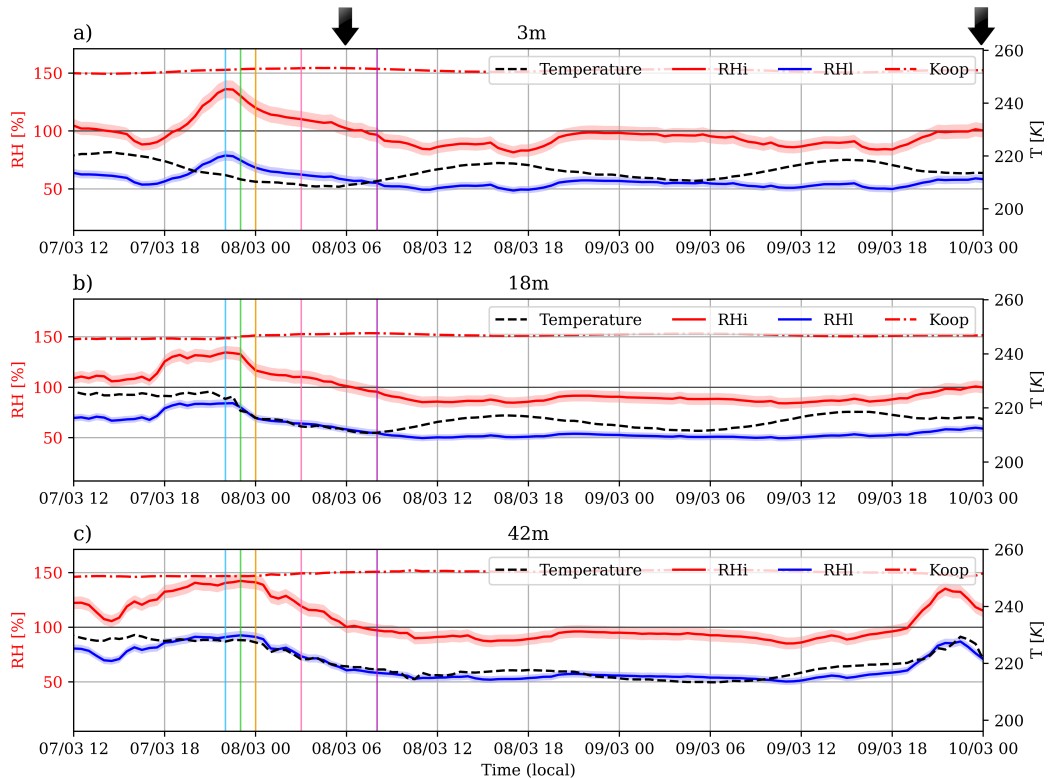

**Figure 4.** Time series of the temperature (dashed black line) and relative humidity with respect to ice (RHi, solid red line) and liquid water (RHl, solid blue line) at 3 m a.g.l (a), 18 m a.g.l. (b) and 42 m a.g.l. (c). The red and blue shading show the uncertainty ranges as estimated in Appendix A. The dotted-dashed red line shows the homogeneous freezing RHi threshold value from Koop et al. (2000). Vertical colored bars indicate the times at which vertical profiles are analysed in Fig. 5. Black arrows indicate the times at which the fog visually appears and disappears near the surface in the lidar measurements.

$7 \, \mathrm{m \, s^{-1}}$. Assuming a logarithmic wind speed profile between the surface and 10-m and an aerodynamic roughness length value
of 1 mm (Vignon et al., 2016), this corresponds to a 3-m wind speed threshold value of $3.5 \, \mathrm{m \, s^{-1}}$. Note that at 2200 LT, 7 March and after 0600 LT, 8 March, the 3-m wind speed exceeds the threshold.

As the air is coming from the southern sector, we do not expect this fog to be associated with an oceanic intrusion. It rather corresponds to a cloud that forms locally or slightly upstream over the Antarctic Plateau. This is confirmed by the analysis of air mass back-trajectories arriving at 50 and 100 m a.g.l. shown in Fig.3a. As the temperature along the trajectories never
exceeds 235 K (not shown), an advection of supercooled liquid droplets towards Dome C is unlikely which confirms that the fog does not have a liquid origin

Temperature measurements on the mast remain below 235 K throughout the event and one can notice a weak diurnal cycle at 3-m that is almost totally damped at 42-m (Fig. 4). A close investigation of the vertical profile of the potential temperature - very close to absolute temperature over the mast depth - reveals that the boundary layer transits between a weakly convective




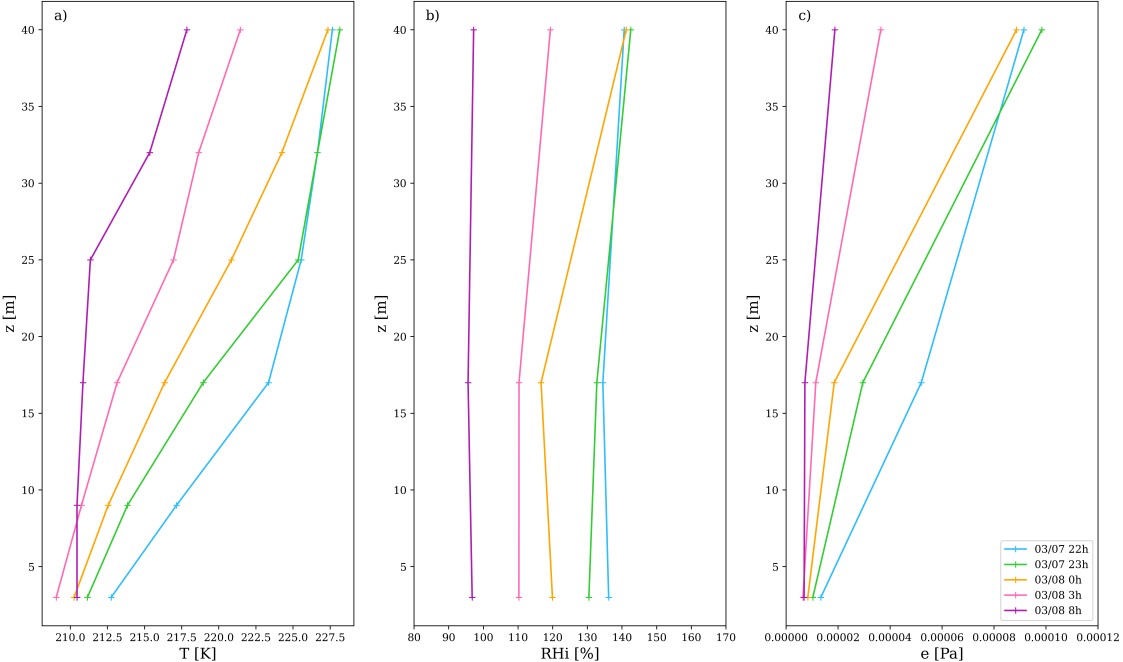

**Figure 5.** Vertical profiles of temperature (a), relative humidity with respect to ice (b) and vapor partial pressure (c) at different times during the fog development phase.

state during daytime to a stable state during nighttime. The fog does not have a clear radiative signature at the surface even though a slight decrease in downward longwave radiation is noticeable during the evening of 7 March (Fig B2a), which concurs with the overall decrease in air temperature (Fig 4).

### 3.1.2 Fog initiation: 1500 LT, 7 March to 0300 LT, 8 March

We now focus on the initiation period from 1500 LT, 7 March to 0300 LT, 8 March. At the 42-m level, RHi first exhibits an increase from 105 % to 142 % which coincides with an increase in vapor partial pressure. At the 3-m level, a later but sharper increase in RHi up to 136 % at 2200 LT, 7 March is noticeable and unlike at 42-m and 18-m, this increase is partly explained by a decrease in temperature during the day-night transition. The maximum RHi reached at the three levels ranges between 136 and 142 %. RHl remains lower than 75 % at 3-m but it exceeds 90 % at 42 m at 2300 LT, 7 March.

RHi then gradually decreases and reaches 100 % at the three levels at 0800 LT (Fig. 4). The decrease starts at 2230 LT at the 3-m level, a half-hour later at 18-m and 2 hours later at 42-m.

At 3-m, RHl and RHi are too low for aerosol deliquescence and solution droplet freezing to occur (Baumgartner et al., 2022). Note that we also ensured that the station exhausts do not affect our measurements (see grey shading in Fig. 3c). We thus expect the decrease in 3-m relative humidity from 2200 LT to be due to a downward turbulent moisture flux towards the





ground surface as the vertical gradient of vapor partial pressure is positive (see Fig. 5c) and/or to the vapour deposition of
drifting ice crystals in the surface layer although the erosion threshold is barely attained.

At 18-m, the decrease in RHi from 2200 LT is co-incidental with a sharp decrease in vapour partial pressure as well as a
decrease in gradient of vapour partial pressure (Fig. 5c) between 3 and 18-m. The 18-m drying from 2200 LT is therefore due
to the turbulent mixing near the suface with a net flux of a moisture oriented downward.

At 42-m, a similar decrease in relative humidity and temperature is observed but the particularity of this level, and probably
that of the overlying air layers is that RHi gets very close to the Koop et al. (2000)'s RHi threshold for homogeneous freezing of
solution droplets in the middle of the night (see time series in Fig. 4c and the how the system evolves in a RH-temperature space
in B1). We can therefore assume that the ice fog forms a few tens of meters above the ground surface through homogeneous
nucleation and the subsequent decrease in RHi at 42-m is therefore partly explained by the vapour deposition onto newly
formed ice crystals. A few patches of ice fogs are noticeable during the night between the 7 and 8 March (Fig. 2a).

### 3.1.3   Growth and decay: 0300 LT, 8 March to 2300 LT, 9 March

From 0300 to 0800 LT, 8 March, the temperature vertical profile shows a clear night-day transition i.e. a transition from an
'exponential' shape characteristic of very stable boundary layers to a convective profile with a well-mixed layer capped by a
shallow temperature inversion whose height further increases during the day (Fig. 5a). A close inspection of the vertical profile
of specific humidity between 3-m and the surface at 0800 LT (not shown) - assuming that the specific humidity at the snow
surface equals the saturation specific humidity at the surface temperature - reveals that the vertical gradient of specific humidity
and subsequently the surface flux of water vapour reverses sign and become oriented upward. The supply of water vapour from
the snow surface - and possibly of drifting ice crystals since the surface wind speed exceeds the erosion threshold (Fig. 3b) -
can therefore deposit onto the ice crystal embryos nucleated in the early morning and make them grow and become visible in
the lidar signal.

Fig. 2a shows that the depth of the fog layer gradually increases from 0600 LT, 8 March up to about 80 m at 1800 LT, 8
March, as the daytime convective boundary layer deepens in $\approx \sqrt{t}$ (Stull, 1990). The growth of the fog is possible as the top
of the boundary layer is supersaturated (Fig 2c) and ice crystals can hence grow by vapour deposition and sediment down to
the subsaturated near-surface layers (Fig. 2c and 4). One can then note a sharper deepening of the fog up to 200 m as night
falls. The mechanisms responsible for this sudden growth cannot be directly identified from our measurements but one can
nonetheless make some assumptions. As the wind speed increases in the late afternoon (Fig. 3b), one could expect an enhanced
vertical mixing of ice particle but a close examination of the wind speed profile from the radiosonde reveals a weak local wind
shear at the top of the boundary-layer. Nonetheless, in the late afternoon, the decrease in the shallow convection intensity leads
to a decrease in the capping temperature inversion strength (Ricaud et al., 2012). The weak vertical gradient of temperature at
the top of the evening convective boundary layer is visible in Fig. 2b (see black arrow) whereas a second higher and stronger
temperature inversion is visible at the top of the fog layer and probably induced by the cloud-top radiative cooling. Izett and
van de Wiel (2020) show that the growth of a radiative liquid fog layer can suddenly accelerate when the vertical gradient
of temperature decreases (i.e. when the vertical gradient of saturation specific humidity decreases, see their Eq. 7). If we as-





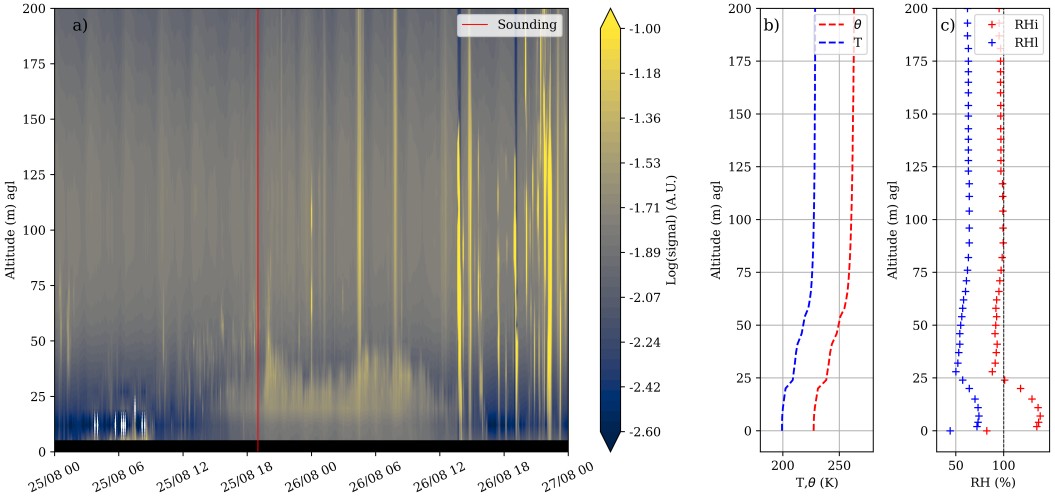

**Figure 6.** Same as Fig. 2 but for Event 2.

sume a homogeneous freezing process, the ice nucleation occurs at the Koop et al. (2000)'s threshold, that is, according to an increasing function of temperature. Subsequently, we can reasonably think that the conceptual model of Izett and van de Wiel (2020) is qualitatively valid for our ice fogs. We can therefore assume that during the evening of the 8 March, the ice fog grows radiatively and the depth of the layer sharply increases due to the decrease in the temperature vertical gradient. It is worth noting that the growth rate of the fog layer (about 120 m in 7 h from visual inspection of Fig. 2a, that is $\approx 17\,\mathrm{m\,h^{-1}}$) is quite similar to the growth rate of the radiative liquid fog shown in Izett and van de Wiel (2020) during the acceleration phase.

The fog grows until it reaches the top of the atmospheric layer supersaturated with respect to ice. From 1800 LT, 8 March to 0600 LT, 9 March, the RHi at the three levels is $\leq 100$ % (Fig. 4) - in agreement with the radiosounding shown in Fig. 2c - while the lidar detects ice condensates from the surface up to 200 m a.g.l. The top-height of the fog layer then drops down to about 50 m at 0700 LT, 9 March and then slightly increases back at a time which correspond to the deepening of the shallow-convective diurnal boundary layer. The fog layer then becomes shallower and shallower from 1500 LT, 9 March as the turbulent mixing in the boundary layer weakens. The fog vanishes during the evening of the 9 March and RHi and temperature at 42-m increase up to their values before fog formation.

### 3.2 Event 2: 25-27 August 2018

#### 3.2.1 Overview

The second case study occurred between the 25 and 27 August 2018. No picture is available since the sun is almost always below the horizon during this period of the year. At 1500 LT, 25 August a very shallow - about 20 m deep - ice fog starts to be



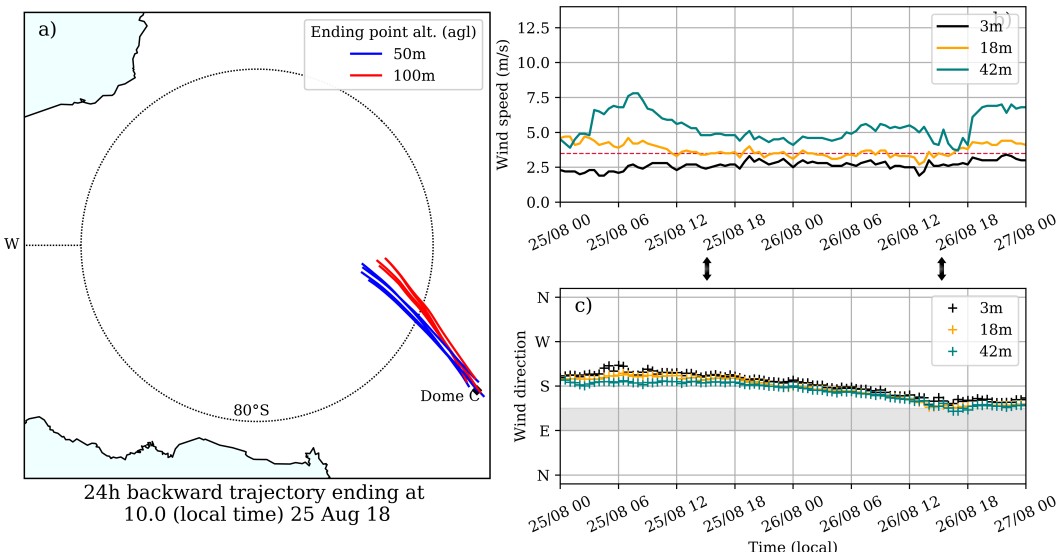

**Figure 7.** Same as Fig. 3 but for Event 2.

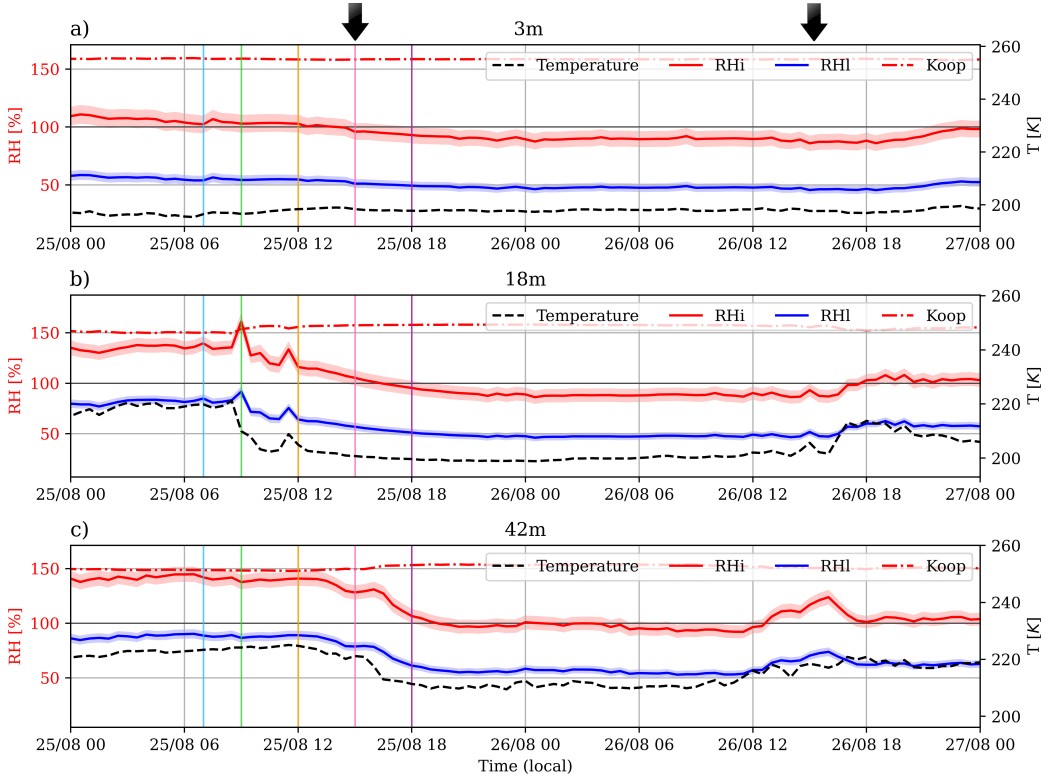

**Figure 8.** Same as Fig. 4 but for Event 2.



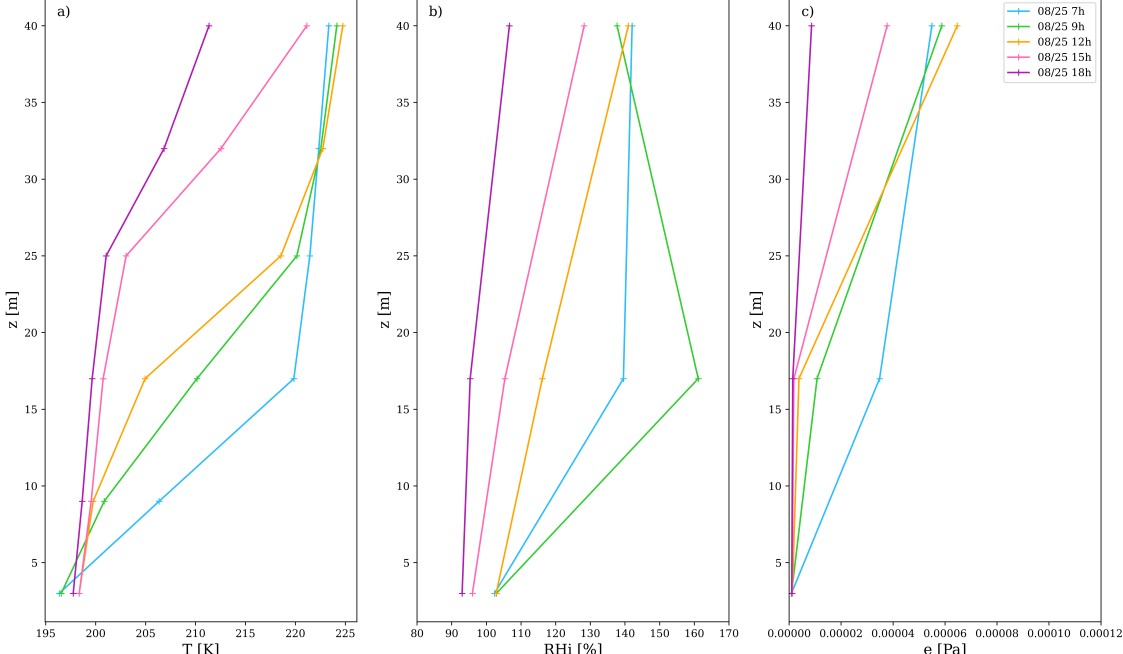

**Figure 9.** Same as Fig. 5 but for Event 2.

visible in the lidar profiles (Fig. 6a). Its depth then oscillates between 25 and 50 m and the fog vanishes at around 1400 LT, 26 August. This event is particularly interesting since its vertical extent is almost the same as the one of the meteorological mast. Note that highly reflective bands are observed during the 26 August and most of them correspond to diamond dust precipitation 'peaks' falling from the mid-troposphere. The radiosounding reveals a supersaturated layer with respect to ice below 30 m a.g.l.

and the relative humidity with respect to liquid water reaches 75% near the surface (Fig. 6c).

Fig. 7b shows that the 3-m wind speed remains below $4\,\mathrm{m\,s^{-1}}$ throughout the event and never exceeds the threshold value to trigger drifting snow. Its direction slightly changes from south-westerly to a south-easterly suggesting that the air parcels reaching Dome C come from the high Antarctic Plateau (Fig. 7b) and that our measurement are not affected by station exhausts except during the very end of the event (see grey shading in Fig. 7c). The back-trajectories shown in Fig. 7a confirm the

continental origin of the flow and as the temperature measured at the station is much lower than 235 K, a liquid-origin of the fog can be ruled out.

The time-series of observational data along the mast reveal a near-constant 3-m temperature of about 200 K during the event and confirm the absence of diurnal cycle during this period of the year.



### 3.2.2   Fog initiation: 0600 to 1500 LT, 25 August

A significant drop in temperature at 0900 LT is observed at 18-m. This cooling is due to an abrupt transition between a very stable to a weakly stable dynamical regime of the boundary-layer (van de Wiel et al., 2017; Baas et al., 2019; van der Linden et al., 2019) driven by an increase in the synoptic wind speed (note that the wind at 42-m, i.e. at the top or above the shallow boundary layer, is a reasonable proxy of the synoptic wind, see Fig. 8c). Subsequently the temperature profile evolves from a typical 'exponential' shape to a concave shape (Vignon et al., 2017) owing to a more vigorous turbulent transport of heat

towards the surface (Fig. 9). At 18-m the cooling rate is particularly intense and equals $5 \text{ K h}^{-1}$ between 0800 and 1100 LT, 25 August. As an analogy with typical air parcel cooling within updraft in the mid- and high-troposphere, such a cooling rate roughly corresponds to an adiabatic air ascent of $0.10 \text{ m s}^{-1}$.

As temperature drops, RHi at 18-m suddenly increases and slightly exceeds Koop et al. (2000)'s homogeneous freezing threshold at 0900, 25 August. RHl exceeds 90% at the same time (see Fig. 8b and B1). The relative humidity does not substan-

tially change at the two other levels, leading to a well marked maximum in the vertical profile (Fig. 9b). It is worth mentioning that Genthon et al. (2022) show that the climatological RHi along the mast also exhibits a clear maximum at 18-m due to non-linear dependence of relative humidity on temperature and atmospheric water content.

Therefore, the initiation of the second fog event is likely due to a homogeneous freezing process starting between 3 and 42 m a.g.l. The density and size of ice crystals become sufficiently large for being well visible in the lidar signal a few hours later.

At 0900 LT, 25 August, RHi at 18-m starts a pronounced decrease due to the vapour deposition onto newly-formed crystals and to a downward turbulent flux of water vapour as the vertical gradient of vapour partial pressure between 3 and 18 m is positive (Fig. 9c).

### 3.2.3   Growth and decay: 1500 LT, 25 August to 1500 LT, 26 August

The decrease in RHi at 42-m occurs at 1500 LT and it coincides with a decrease in temperature associated with the deepening

of the weakly stable boundary layer (Fig. 8c and 9a). The decrease in the 42-m RHi also concurs with the increase in fog layer depth visible in the lidar data between 1500 and 1800 LT, 25 August (Fig. 6a) and can therefore be attributed to the deposition of water vapor onto ice crystals. The saturation is reached at 1900 LT, 25 August.

At 3-m, RHi remains at, or slightly below, saturation during the period of the fog suggesting that ice crystals at this height do not neither nucleate or grow but sediment from higher layers.

At 1200 LT, 26 August, the fog starts to dissipate from the top and the air at 42-m becomes supersaturated wrt ice (Fig. 8c). A few hours later a decrease in RHi back to saturation occurs probably due to vapor deposition onto the diamond dust preicpitation streaks that suddenly fall down from the low- and mid- troposphere (Fig. 6a).



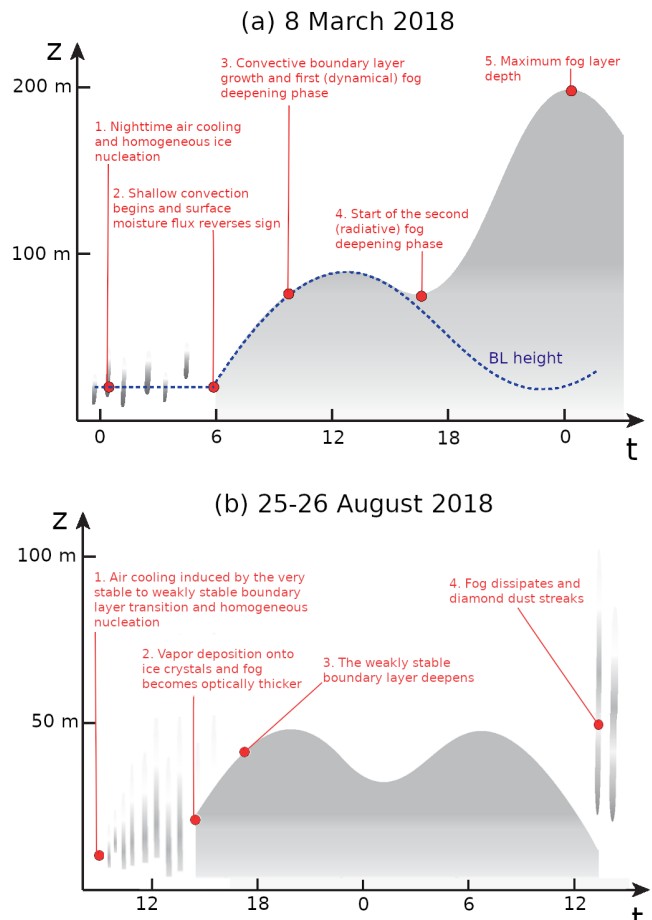

**Figure 10.** Conceptual scheme of the formation and growth of the two ice fogs studied in the paper.

## 4 Summary and conclusions

Temperature and humidity measurements from advanced thermohygrometers along a 45-m mast are combined with lidar and
radiosonde observations and air-parcel back-trajectories to study the development of two fog events at cirrus temperatures in the
near-surface atmosphere of Dome C, high Antarctic Plateau. For both case studies, fog forms locally and does not correspond
to warm maritime intrusions. Lidar observations evidenced shallow fogs and the innovative moisture measurements along the
tower allowed us to explain the mechanisms responsible for their formation.

Fig 10 summarises the development of the fogs with two schematics. The values of supersaturation that are attained and the
very low temperatures that prevent the pre-existence of supercooled liquid droplets suggest that the two fogs are initiated by the
homogeneous freezing of solution aerosols in the stably stratified boundary layer. To our knowledge, this our study presents the
first observational evidences that this process can be responsible for the formation of an ice fog. The highest supersaturations



occur at a height where the air cools down by turbulent mixing and radiation and where the downward flux of water vapour is sufficiently weak. The evolution of the fog is thereby tightly related with the dynamics of the boundary-layer which experiences a weak diurnal cycle in the first study case and a dynamical transition between a very stable and a weakly stable state in the second case.

Regarding the potential similarity between Antarctic ice fogs and cirrus clouds we raised in the Introduction, this study suggests that the homogeneous freezing of solution particles, i.e. a common path to cirrus cloud formation, can be studied in natural conditions near the ground surface of the Antarctic Plateau. More generally, it emphasises that Dome C is a relevant place to carry out observational studies of microphysical processes in ice clouds. Furthermore, the analysis of humidity measurements during the growing phase of fog gives access to the time-scale at which the vapour is depleted, even though it is particularly delicate to disentangle the dynamical - i.e. turbulent mixing - from the microphysical - i.e. deposition onto ice crystals - causes. The development of the fogs is indeed tightly coupled with the dynamics of the boundary-layer. This is a non-negligible difference with cirrus clouds even if the dynamics thereof can be very turbulent (e.g., Gultepe and Starr, 1995).

While the available observations at Dome C make it possible to characterise the overall development of ice fogs, they do not give direct information about the type - homogeneous or heterogeneous - of the ice nucleation process and do not allow for a fine understanding of the interactions between cloud microphysics, radiation and turbulent dynamics. Collecting sedimenting ice crystals during ice fog events and establishing formvar replicas thereof in the manner of Santachiara et al. (2016) would allow us to analyse the morphological structure of crystals and to perform chemical analyses of potentially remaining particles after sublimation. Running a large-eddy simulations with an advanced microphysical scheme for cold clouds and capable of simulating the boundary-layer at Dome C (Couvreux et al., 2020) would also help better understand the mechanisms driving the growth and decay of the fogs. Furthermore, investigating the frequency of occurrence of very cold ice fogs at Dome C was beyond the scope of this study. This aspect would deserve further attention in the future to figure out their climatological impacts on the Antarctic Plateau.





**Appendix A: Uncertainties of relative humidity measurements**

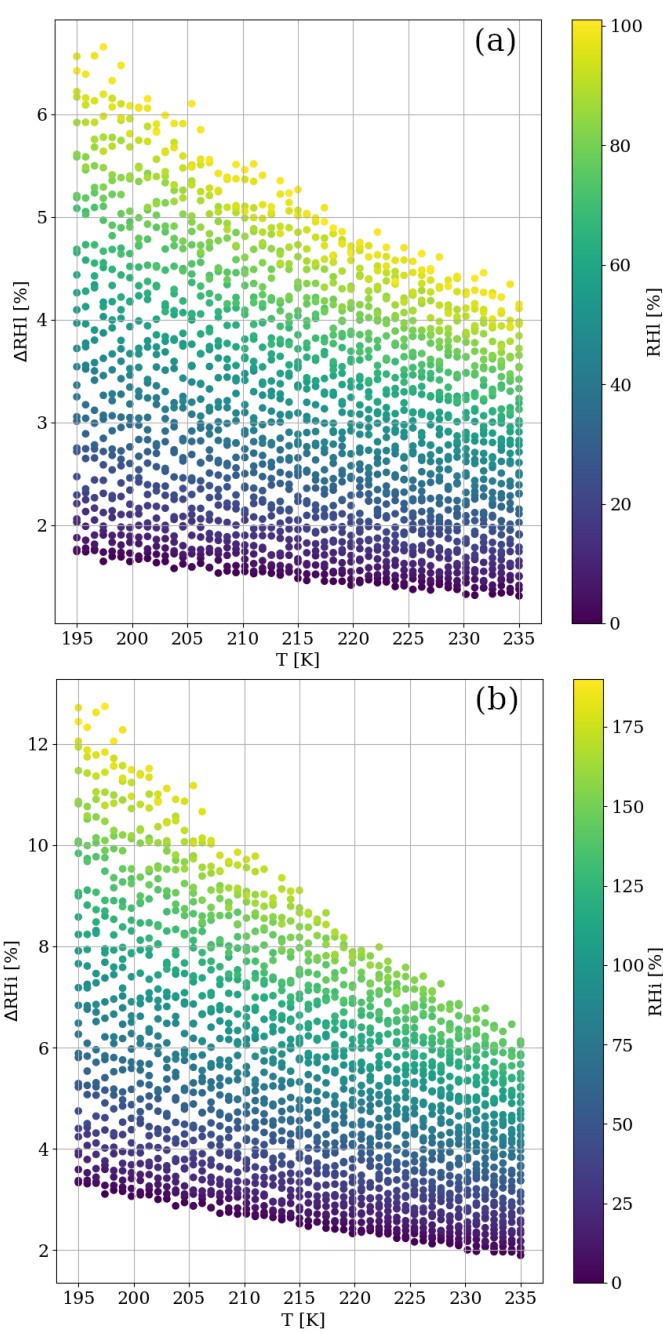

**Figure A1.** Panel a (resp. b): uncertainty of the RHl (resp. RHi) estimations as a function of temperature (x-axis) and RHl (resp. RHi) (color shading).




The estimation of RHl and RHl from the advanced thermo-hygrometers depends on (Genthon et al., 2017, 2022):

– the measurement of the relative humidity wrt liquid water in the heated inlet $RHl_h$ by the HMP-155 probe;

– the measurement of the temperature of the heated inlet $T_h$ by the HMP-155 (PT-100);

– the measurement of the ambient temperature $T$ by the independent PT100 platinium resistance.

RHl and RHi are then calculated with the following expressions:

$$RHl = RHl_h * e_{sl}(T_h)/e_{sl}(T) \tag{A1}$$

$$RHi = RHl * e_{sl}(T)/e_{si}(T) \tag{A2}$$

with $e_{sl}$ and $e_{si}$ the saturation water pressure with respect to liquid and ice respectively and calculated with Murphy and Koop (2005)'s formulae. According to the manufacturers, the accuracy of the relative humidity measurement below $-40^{\circ}$C is

$\pm 1.4 + 0.032$ of the reading in percentage and the accuracy of the temperature measurement is $0.226 - 0.0028 * (T - 273.15)$. To estimate the uncertainties of the RHl and RHi end-products, we perform a Monte Carlo test with a 1'000 uniform re-sample of $RHl_h$, $T_h$ and $T$ in the interval [value - accuracy,value + accuracy]. The uncertainties $\Delta$RHl and $\Delta$RHi are evaluated as one standard deviation of the obtained distributions for each bin of relative humidity and temperature. In the calculation, we assume that the air reaching the hygrometer sensor in the heated inlet is $\approx 5$ K warmer than the ambient air (mean over the

measurement period: 4.9 K, standard deviation: 0.7 K).

Panel a (resp. b) of Fig. A1 shows the dependence of $\Delta$RHl (resp. $\Delta$RHi) to temperature and RHl (resp. RHi). $\Delta$RHl and $\Delta$RHi can be expressed as continuous functions of relative humidity and temperature with a regression of the form:

$$\Delta RHl, i = a0_{l,i} + a1_{l,i} RHl, i(a2_{l,i} T + a3_{l,i} T^2) \tag{A3}$$

with $a0_{l,i}$, $a1_{l,i}$, $a2_{l,i}$ and $a3_{l,i}$ the regression coefficients. The $R^2$ coefficients of the regression equal 0.993 and 0.991 for

$\Delta$RHl and $\Delta$RHi respectively.





## Appendix B: Additional figures

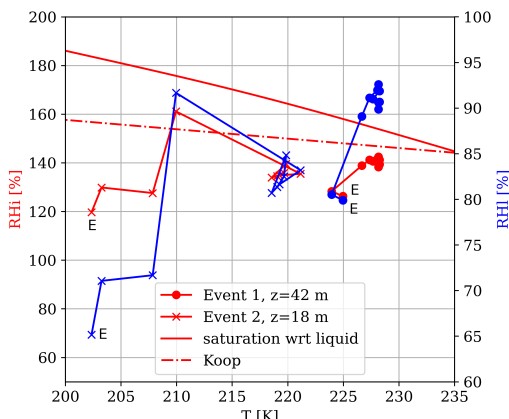

**Figure B1.** RHi (red curves, left y-axis) and RHl (blue curve, right y-axis) evolution as a function of temperature at the level of nucleation during the initiation phase of the fog. Lines with circles refer to the first event (from 2100 LT, 7 March to 0200 LT, 8 March) while the lines with crosses refer to the second event (from 0600 to 1100 LT, 25 August). 'E' indicates the endpoint. The solid red line shows RHi value at liquid water saturation and the dotted-solid shows the Koop et al. (2000)'s homogeneous nucleation threshold.

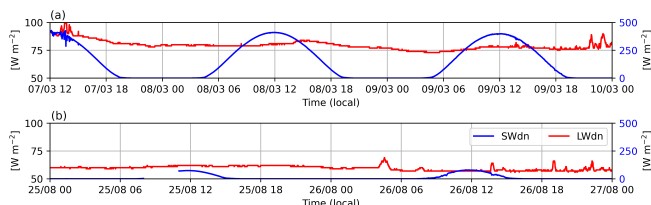

**Figure B2.** Time series of downward longwave (LWdn, red) and shortwave (SWdn, blue) radiative fluxes at the surface during the first (a) and second (b) case studies.



*Data availability.* The Dome C temperature, wind and humidity datasets used in this study are thoroughly described in Genthon et al. (2021, 2022) and distributed on PANGAEA (Genthon et al., 2021a, b, 2022). Radiosonde data are freely distributed at www.climantartide.it. The tropospheric depolarization lidar data can be obtained at http://lidarmax.altervista.org/englidar/_AntarcticLIDAR.php. BSRN data are
available on PANGAEA (Driemel et al., 2018)

*Author contributions.* EV, LR and CG designed and conducted the study. LR and EV analysed the data and LR made the figures. MDG collected and processed the lidar data. CG collected and maintained the meteorological measurements on the mast at Dome C. AH, JBM and AB provided scientific expertise on cold microphysics and contributed to the results' interpretation. EV wrote the paper with contributions from all the authors.

*Competing interests.* The authors declare they have no competing interests.

*Acknowledgements.* We gratefully thank Roxanne Jacob for examining additional photographs during fog events at Dome C. This research was conducted in the framework of the CALVA 1013 and PRE-REC observation programs with the support of the french polar institute (IPEV) and the Programma Nazionale di Ricerche in Antartide (PNRA). Radiosounding data have been acquired from the database of the IPEV/PNRA project 'Routine Meteorological Observation at Station Concordia' www.climantartide.it with the help of Paolo Grigioni. This
project has received funding from the European Research Council (ERC) under the European Union's Horizon 2020 research and innovation programme (grant agreement No 951596).



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
