# Peer review of "Ice fog observed at cirrus temperatures at Dome C, Antarctic Plateau"

_EGUsphere, 2022_

## Author Comment (AC1)

**Response to Review #1**

The authors report on the observation of two cases of ice fogs formed at Dome C, Antarctica. Both ice fogs formed at very cold temperatures which are typical for cirrus clouds in the upper troposphere. In particular, these fogs formed in-situ, most likely due to the homogeneous nucleation of ice crystals, i.e. the freezing of pre-existing aerosol solution particles. This aspect makes this study particularly interesting since in contrast to aircraft based observations of natural (i.e. outside of the laboratory) cirrus clouds the authors are able to show timeseries of key parameters at a stationary location, hence within the forming cloud itself.

The manuscript is very well written and fits well into the scope of ACP. After adressing my comments and questions I have listed below, I recommend the acceptance of the manuscript.

We gratefully thank Reviewer #1 for his/her review our paper and for recommanding its acceptance after revision. We answer his/her line by line comments herebelow :

**Comments:**

(1) I find it a pity that no data is presented that can shed some light on the nucleated ice crystals within the two fogs, such as their shape, mass, number density. If there is such data available, I strongly recommend to include that.

We completely agree with you, it is a pity that no measurements of the shape, mass and density of ice crystals at Dome C is available during our two fog events. That is why one prospect of this work is to collect sedimenting ice crystals during ice fog events and establish formvar replicas in the manner of Santachiara et al. (2016). Note that an 'ice camera' has been operating for several years at Concordia : http://lidarmax.altervista.org/englidar/Antarctic%20Precipitation.php

It takes pitures of snowflakes and an algorithm classifies them according to their size and geometrical properties (see an example in the figure below). While the camera provides interesting information during synoptic precipitation event, no crystal photograph was available during the two fog events because the size of the tiny fog particles was below the detection limit of the instrument.

(2) Line 21-22: You state that the ice crystal properties "such as their size and their number concentration" are different for an ice fog or diamond dust. I suggest adding a sentence to clarify these differences.

We have added the following sentences :

'Ice fogs can be distinguished from another surface ice cloud type called `diamond dust' through visibility criteria - fogs are optically deeper - or ice crystal properties. For instance, Girard & Blanchet (2001) distinguish ice fog from diamond dust by the high concentration of ice crystals of small diameters as the particles' number concentration in fog clouds generally exceeds 1000 L-1 and their size is below 30  $\mu$ m.'

(3) Line 30-32: To me it seems that the formation process you describe here is the freezing of supercooled liquid droplets which are already as large as cloud droplets. In other words it is the freezing of a pre-existing liquid cloud. I suggest to refer to this process as a liquid-origin cloud, since the term "homogeneous freezing" is usually understood as the freezing of much smaller solution aerosol particles (you describe this process in line 41-44).

You are right we describe here the main formation processes described in Gultepe et al. (2017) which correspond to liquid-origin fogs. We have added 'liquid origin' where it is relevant in the text.

(4) Section 2.2: At very cold temperatures close to 200K, a new formulation of the saturation vapor pressure over liquid water was recently presented by Nachbar et al. This formulation differs from the formulation given in Murphy and Koop (2005), in particular at cold temperatures. What happens to your RHl-values if you use this new formulation? Although Nachbar et al state that their parameterization is only valid for temperatures above 200K, it seems that such a comparison is applicable for observed fog in the case 1. Of course, such a comparison might also affect the results in Appendix A.

**Reference:**

Nachbar, M., Duft, D., and Leisner, T.: The vapor pressure of liquid and solid water phases at conditions relevant to the atmosphere, J. Chem. Phys., 151, 064504, https://doi.org/10.1063/1.5100364, 2019.

Thank you very much for raising this point and for refering to this paper we did not know. We have calculated RHI following Nachbar et al. (2019) and compared the results with the calculations using Murphy and Koop's esl formula.

The first figure herebelow compares the RHl estimations at the three levels and over the whole year 2018. One can notice that the two formulae overall give similar results.

The second and third figures shows the difference in RHl and RHi using the two formulae (at the three levels and over the whole year 2018) versus the ambient air temperature. Differences are almost always lower than 1.2 % (in magnitude) for RHl but reach a few percents at very low temperatures for RHi (as the difference in esl and esi is very small). Even if those small RH differences do not affect our overall results and conclusions regarding the fog initiation and

evolution, we want to show the most accurate RH estimations as possible. Following your recommendation, we have therefore recalculated the RHI and RHi value using Nachbar's esl formula, recomputed the observational uncertainties in Appendix A and updated the figures 2,3,4,5,8,9,A0 and A1 (and the numerical RHI and RHi values given in the paper).

(5) Section 2.5: Does the Global Data Assimilation System employ a rotated grid to avoid a polesingularity in the Antarctica area? If not, does this singularity affects data that is used to compute the backward trajectories?

The Global Data Assimilation System is based on the Global Forecast System model which uses the Finite Volume Cubed Sphere (FV3) dynamical core. The latter is a finite-volume core with no singularity at the poles :

https://www.gfdl.noaa.gov/fv3/fv3-grids/

Our back-trajectories are thefore not affected by singularity-related effects.

(6) Figure 4 and 8: I suggest to indicate the two time periods which you describe in the following subsections with "Initiation" and "Growth+Decay" within the figures, e.g. by adding two vertical arrows below the panels. In addition, I suggest to indicate the time which corresponds to each of the vertical lines shown in the panels (e.g. by adding the times at the top of the first panel). What is the meaning of the solid black horizontal line?

Thank you for this comment. We have modified Figures 4 and 8 with your suggestions. We also specify in the caption that the solid black line highlights the 100 % RHi value.

(7) Line 151: Note that Baumgartner et al (2022) describe that the homogeneous freezing of the solution particles already starts at values of RHi below the threshold given in Koop et al (2000). The rate of ice crystal nucleation increases as the values of RHi approach that threshold, but the threshold is not to be understood as a switch. In essence, as long as RHi comes close to the critical value (e.g. the threshold), the homogeneous nucleation starts and there might have been some homogeneous nucleation also at 3m height during your observation.

Thank you very much for raising this point. We agree that the Koop's threshold should not be interpreted as a binary switch. First of all, following the review by the editor M. Krämer, we have added a  $\pm$  5 % enveloppe around the Koop's curve in our graph (see the explanation in our response to her comments). Moreover and following your recommendation, we have reformulated the paragraphs analysing situations for which RHi is below but close to the Koop et al (2000)'s threshold :

For the first event :

'Given the maximum RHI and RHi values attained, the aerosol deliquescence and solution droplet freezing at 3-m a.g.l. are not very likely but their occurrence cannot be completely excluded since Baumgartner et al. (2022) show that homogeneous freezing can start at RHi values slightly lower than the Koop et al. (2020)'s threshold.'

For the second event :

'At 42-m, RHi approaches the Koop et al (2000)'s threshold between 0400 and 0500 LT 25 August and some preliminary crystal nucleation can already occur at this time.'

Minor and technial comments:

(1) Line 27: It should read "pre-conditioned"

Corrected.

(2) Line 59: "...data at Dome C, a site particularly..."

Corrected.

(3) Line 104: "droplets"

Corrected.

(4) Line 114: "to track the trajectories of the air masses probed above Dome C."

**Corrected.**

(5) Line 120: It should read "0600 LT" ?

Yes indeed. This has been corrected.

(6) Line 149: It should read "0800 LT, 8 March (Fig. 4)." and "2230, 7 March, at the"

Thank you. This has been corrected.

(7) Line 209-210: What is the maximum value of Rhi?

The maximum RHi value attained along the radio sounding is 113 % . We have added this information in the text.

(8) Line 213: "measurements"

Corrected.

(9) Line 247: "precipitation"

Corrected.

(10) Line 256: Delete "this"

Done.

(11) Appendix A: I found it quite hard to understand what exactly is shown in figure A1. Please state this more explicitly. It would also be helpful to add a sentence on how one should "read" these plots.

Following your suggestion, we have added the following sentence :

'Panel a (resp. b) of Fig A1 shows how the uncertainty in the RHl (resp. RHi) estimation depends on temperature. For each temperature bin on the x-axis, the dependence is explored for different RHl (resp. RHi) values below liquid saturation (color shading).'

(12) Line 281: It should read "RHl and Rhi"

Corrected.

(13) Equations A1, A2 and line 290: Please substitute the asterisk by a centered dot to indicate multiplication.

**Done.**

(14) Equation A2: The numbers of the regression coefficients should appear as an index.

Done.

---

## Author Comment (AC2)

**Response to Review #2**

This paper reviews two fog cases that illustrate some unique fog formation means that have likely not been observed or well studied to date. This research is worthy of publication. There are a few key elements that of which some are critical to denote:

We gratefully thank the reviewer for the careful evaluation of our manuscript and constructive comments.

Review:

Line 35 - Supercooled liquid has been observed to 240 K at South Pole Station during the SPARCLE experiment (see https://agupubs.onlinelibrary.wiley.com/doi/epdf/10.1029/2021JD035182)…While this was at a higher altitude above the ground, it is possible to have cold temperatures and still have liquid… Hence, this should be considered in what is written here to denote this. Please revise the temperature values at which liquid water can and does exist.

Thank you very much for mentioning this very interesting paper. We have reformulated the corresponding sentence and refer to Rowe et al. 2022 :

*Given the low concentrations of INPs over the Antarctic (Belosi et al., 2014), supercooled liquid droplets have been observed at very cold temperatures down to 240 K (Silber et al., 2019; Ricaud et al., 2020; Rowe et al., 2022) and they were shown to be at the origin of fogs over the Antarctic coast (Kikuchi, 1971, 1972).*

Line 70 - Averaging data over 30 minutes is a long time. This reviewer is not in favor of this practice as you are smoothing out the data before analysis….which this is less of an issue with slower changing parameters like pressure (not used in this study) but it has a larger impact on faster moving variables such as temperature and wind. As a note, 30 minute averaging is at least 3 times beyond the WMO recommendations which recommend averaging over small time frames (1 minute for temperature, 2 minute or 10 minute for wind) see WMO Publication #8). While we can debate the merits of this, I wonder the impact it would have in interpreting the 25 August case between 6 and 15 LT when the RHi goes above the Koop et al (2000) value. How might the data observations look in this time period without the 30 minute averaging, but instead 10 minute averaging? This non-standard method for handling the data impacts future comparisons likely to be made by others and other observational datasets that do not do this. This contributes to the heterogeneous observing network Antarctic suffers from, and it is not getting any better with divergent observing schemes that are in place.

Thank you for this very relevant comment. In 2009 when the first instruments were deployed on the 45m mast at Dome C, we chose to save 30-min statistics (mean, min, max and variance, see Genthon et al. 2010). 30 min was a good trade-off between the typical time-scales of processes we wanted to characterize and the storage capabilities at that time. Moreover, it should be underlined that the response time of the humicap sensor of the HMP increases well beyond 1 min at T <60°C (Vaisala, personal communication)  and a 1-min resolution is very likely not adapted for relative

humidity data at low temperature. The temperature, wind and humidity datasets officially published and distributed (Genthon et al. 2021, 2022) thus have a 30-min resolution for consistency of the data format throughout the period of measurements.

Nonetheless, for scientific motivations and to comply with WMO standards, 1-min wind, temperature and humidity data have consistently been saved but only since 2019. Unfortunately, no humidity measurement is available for the period of the second fog event and we are not able to analyse the 25 August fog event using data at higher temporal resolutions. We tried to identify a similar but more recent potential fog event from RHi time series and found one in June 2020. The figure below shows that RHi evolves quite smoothly during the fog formation. Even though a higher time resolution helps identify more accurately the RHi value and the time at which ice crystals nucleate, the general conclusions regarding the evolution of RHi during the fog formation are the same using 10-min and 30-min averages.

[Figure]

*Figure : 18-m RHi time series in June 2020. The red (resp. black) line show 10-min (resp. 30-min) moving averages of 1-min resolution data. The purple line shows the Koop (2000)'s threshold.*

Genthon C, Town MS, Six D, Favier V, Argentini S, Pellegrini A. 2010. Meteorological atmospheric boundary layer measurements and ECMWF analyses during summer at Dome C, Antarctica. J. Geophys. Res. 115: D05104, doi: 10.1029/2009JD012741

Genthon, C., Veron, D., Vignon, E., Six, D., Dufresne, J. L., Madeleine, J.-B., Sultan, E., and Forget, F.: Ten years of wind speed observation on a 45-m tower at Dome C, East Antarctic plateau, https://doi.org/10.1594/PANGAEA.932513, 2021

Genthon, C., Veron, D., Vignon, E., Madeleine, J.-B., and Piard, L.: Water vapor observation in the lower atmospheric boundary layer at Dome C, East Antarctic plateau, https://doi.org/10.1594/PANGAEA.939425, 2022.

Line 80 - As I read through the cases, I wonder if it would help the reader to know more about the Murphy and Koop methodology, as seeing an RH value of over 100% seems unexpected (but it is fine, correct?).

We refer to Murphy and Koop here as they provide state-of-the-art formulae for the saturation vapor pressure with respect to liquid and ice phases. We guess you mean that further details are needed to explain the Koop (2000)'s (nor Murphy and Koop's) approach to explain the homogeneous freezing at high supersaturation values.  Following your recommandation and the one from the editor, we have therefore added the following paragraph in Sect. 2.2 :

*To detect the possible occurrence of homogeneous freezing of solution aerosols, we will compare our RHi measurements with the so-called Koop et al. (2000)'s threshold. In the approach of Koop et al. (2000), solution particles spontaneously freeze when RHi exceeds a threshold value that primarily depends on temperature. As a first approximation, we calculate the RHi threshold value ($RHi_T$ , in %) using the analytical fit of Koop et al. (2000)'s experimental results derived in Ren and Mackenzie (2005):*

$$RHi_T = (2.349 − T/259) \cdot 100$$

*where T is the temperature in Kelvin. This fit has been performed for solution particles in equilibrium with the ambient vapor that have a typical radius of 0.25 µm and that can freeze homogeneously within 1 min (see also Kärcher and Burkhardt, 2008). The exact value of the threshold also depends on the size of the particle as well as on the composition thereof and on theformulation and uncertainties of water activities and saturation vapor pressure. Individually, those effects make $RHi_T$ vary by about 1 to 5 % (see Baumgartner et al., 2022). An envelop of 5 % has therefore been added around the Koop's curve in our graphs. This envelop is only intended as a rough indicator of the uncertainty and to guide the eye.*

Also, the RHi vs. RHl seem to be the same curves with an offset (?) Using RH overall is a terrible measure of actual moisture anyways…and RHi clearly shows that you are saturated or supersaturated with respect to ice.

Thank you for this comment. The ratio between RHi and RHl is a function of temperature only (esl(T)/esi(T)). In Fig. 4 and 8, as temperature does not substiantially vary, the RHl curve looks shifted from RHi but this is physically consistent. We agree with you that RHi is the most informative variable for our work but the examination of RHl informs about the potential degree of deliquescence of aerosols (and possible subsequent homogeneous freezing). This is why we show both RHi and RHl in Figures 4, 8 and B1.

Figure 2 - Is this for the March 8th case?  Some indication of dates/times in the caption would be helpful.

We have reformulated the caption as follows :

*Panel a: Time-height plot of the lidar backscattering signal intensity during the first fog event between 12 LT, 8 March to 00 LT, 10 March 2018. The red line indicates the radiosonde launching time. Panel b shows the vertical profile of temperature (blue) and potential temperature (calculated with a reference pressure of $10^S$ Pa, red) from the radiosounding. The black arrow indicates the top of the boundary layer. Panel c shows the vertical profile of relative humidity with respect to ice (RHi, red) and with respect to liquid water (RHl, blue) from the same radiosounding.*

Figure 3 - So this case, you have wind speeds clearly over the threshold for blowing snow, yet it is not reported nor happening?  (Also see lines 125 through 130…)

To our knowledge, blowing snow has never been reported at Dome C but snow drift does occur when the wind is sufficiently strong. We cannot report the occurrence of drifting snow as the lowest lidar reliable gate is at ~ 20 m a.g.l. i.e at a height much higher than a typical drifting snow layer depth. Moreover, no instrument specifically designed to detect blowing snow such as Flowcapts or SPCs have been deployed at Dome C so far.

A Campbell SR50 acoustic depth gage measures the local variation of the snow surface at Dome C (see Genthon et al. 2015) but the surface footprint of the SR50 is very small and no robust conclusions regarding the occurrence of snowdrift can be drawn from the measurements of a single instrument. We have therefore specified in the text that snow drift is possibe when the wind speed exceeds the threshold value :

*'Note that at 2200 LT, 7 March and after 0600 LT, 8 March, the 3-m wind speed exceeds the threshold and some snow drift is therefore possible during those periods.'*

Genthon, C., D. Six, C. Scarchilli, V. Ciardini, and M. Frezzotti (2015), Meteorological and snow accumulation gradients across Dome C, East Antarctic plateau, Int. J. Climatol., 36, 455–466, doi:10.1002/joc.4362.

Line 120 - Reference Figure 4 here with the RHi value referenced…

This paragraph portrayed the overall evolution of the fog from the lidar measurements. The evolution of the relative humidity from the hygrometers data is thoroughly described in  subsequent paragraphs. To avoid any confusion, we prefer not to refer to Fig. 4 in this paragraph.

Line 150-155 Is it fluxing downward and the atmosphere is not decoupled at all above?? *** Unlikely there is decoupling?***

Thank you for raising this point. Indeed the 3-m level is not decoupled from the atmosphere above. However, the vertical gradient of partial pressure is stronger near the surface, one can therefore reasonably assume that the divergence of the flux at 3-m is positive (leading to a decrease in moisture content). We now specify in the text that we speaking about the 'net' flux.

Line 205 - Is the 20 meters from human observation?

We now specify 'above ground level' in the text.

Line 217 - This stray sentence should be combined with the paragraph above.

Done

Line 243 - This stray sentence should be combined with the paragraph above.

The paragraph has been reformulated to answer a comment from a second reviewer. There is no stray sentence anymore.

Figure 10 is really helpful - just too small. Is there anyway it can be published to be larger to see the red text??

Following your recommandation, we have increased the size of the figure.

Figure B2 is too small to see - hopefully this can be improved in publication

The size of this figure has also been increased.

Minor langage/English:

Line 5 - Remove "To our knowledge" is not really needed… Just say "This is the first time…"

Corrected.

Line 40 - Remove "hitherto" as it is not needed

Corrected.

Line 256 - Correct the line "To our knowledge, this our study presents…" to simply say "This study presents…"

Corrected.

Line 262 - Remove " we raised in the Introduction" as it is not needed

Corrected.

Line 285 - Add "thermometer" at the end of this bullet point.

Added.

---

## Author Comment (AC3)

**Response to comments by Editor Martina Krämer**

**Comments to the author**:
Étienne Vignon and co-authors,

I'm please to accept your very interesting study for publishing in ACPD. I have some questions (listed below), which can be discussed in the open discussion phase of the paper, because at the current stage of the review process (quick access review) only technical issues should be considered. But of course you could take the comments into account already now if you want.

With kind regards, Martina Krämer

Dear Martina Krämer,
Thank you very much for this enthusiastic general comment and for publishing our paper in ACPD. We also thank you for your very constructive comments. Please find herebelow our response.

- How do you avoid ice particles to enter the intake, evaporate and add humidity ?

This point was already raised by a reviewer of the first paper presenting the hygrometers (Genthon et al. 2017, see discussion here :
https://acp.copernicus.org/articles/17/691/2017/acp-17-691-2017-discussion.html
We particularly emphasized that blowing-snow particles and snow flakes are not expected to subtantially impacts the RH measurements. Moreover, we underlined the fact that above -50°C, the RH data from this instrument compares very well with RH measurements from a frost-point hygrometer

Regarding more specifically diamond dust and fog cristals, note that the two inlets from which the air enters the system are oriented downward, which prevent sedimenting particles from directly falling into the instrument. (see Fig 2 of Genthon et al 2017, www.atmos-chem-phys.net/17/691/2017/). The latter is not equipped with any filters or with another system that block the advection of particles from below (due to turbulent motions or because the air is mechanically aspirated by the fan). However in the present study cases, the aspiration of thin-fog particles (if any) is not expected to substantially affect our measurements.

If we assume a fog formed by homogeneous nucleation under the typical cooling rate that are observed, the number concentration of ice particles at a temperature of ~220 K would be of the order of $5 \, 10^{-1} \, cm^{-3}$ (according to Fig 1b in Baumgartner et al. 2022). Let's further assume that the mean particle dimension is around 10 microns (which is the order of magnitude of diamond dust and fog particles collected by Santachiara et al. 2016, see their Figs 1 and 2). One can thereby calculate the tendencies of water vapor partial pressure de and temperature dT (and saturation vapor pressure desl from Clausius Clapeyron's equation) if all condensates sublimate in the intake.

The change in relative humidity wrt liquid RHl (quantity measured by the HMP) associated with the sublimation of fog particles can then be calculated as :
$dRHl=(esl^{-2})(esl \, de - e \, desl)$
The small python code that was used to estimate dRHl is available here :
https://web.lmd.jussieu.fr/~evignon/paper_fog_acp/

For an ambient RHl value of 60 %, our calculations give a dRHl value of 1.8 %.
This value is not negligible but generally lower than the measurement uncertainties given in Appendix A.

It is also worth remembering that we have assumed spherical ice particles and that all of them sublimate in the intake. Those two assumptions lead to an overestimation of the mass of ice that sublimates and the dRHl value given above is therefore probably overestimated.

In the revised version of the paper, we have added the following paragraph to explain this point :

*'It is worth mentioning that the intakes are oriented downward such that sedimenting ice crystals cannot directly fall into the measurement system. Nevertheless, if the instrument is embedded in a foggy air, some ice particles may enter from below due to turbulent eddies and the mechanical aspiration. We have therefore analytically estimated the effect of sublimation of thin-fog ice particles – with a typical size of about 10 mum (Santachiara et al. (2016)) and a typical number concentration of 0.5 $cm^{-3}$ (Baumgartner et al. 2022) - on the RHl measurements (not shown). The obtained RHl change values are of the order of a few percents at most depending on the ice crystal shape assumption, on the ambient temperature and relative humidity values, and on the fraction of particles that sublimate. In any case, the obtained values are lower than the instrumental uncertainties calculated in Appendix A. The sublimation effects can therefore be deemed second order with respect to the intrinsic temperature and RHl measurement uncertainties.'*

Baumgartner, M., Rolf, C., Grooß, J.-U., Schneider, J., Schorr, T., Möhler, O., Spichtinger, P., and Krämer, M.: New investigations on homogeneous ice nucleation: the effects of water activity and water saturation formulations, Atmospheric Chemistry and Physics, 22, 65–91, https://doi.org/10.5194/acp-22-65-2022, 2022.

Genthon, C., Piard, L., Vignon, E., Madeleine, J.-B., Casado, M., and Gallée, H.: Atmospheric moisture supersaturation in the near-surface atmosphere at Dome C, Antarctic Plateau, Atmos Chem Phys, 17, 1–14, doi:10.5194/acp-17-1-2017, 2017.

Santachiara, G., Belosi, F., and Prodi, F.: Ice crystal precipitation at Dome C site (East Antarctica), Atmospheric Research, 167,108–117, https://doi.org/https://doi.org/10.1016/j.atmosres.2015.08.006, 2016

- A similar instrument as yours to measure humidity exist in an airborne version, here is a reference in case you want to cite it:
https://amt.copernicus.org/articles/8/1233/2015/ and here you can see where it is operated:
https://www.iagos.org/iagos-core-instruments/

Thank you for mentioning this airborne instrument that we did not know. The design and the concept of the their instrument and ours are not exactly the same but in both cases the relative humidity has to be estimated from a relative humidity measurement at a 'sensor' temperature that differs from the ambient air temperature. We have added a reference to Neis et al. (2015) in our paper.

- How did you calculate the homogeneous freezing threshold? Explicitly or as an approximation?

Thank you for raising this point that we have not detailed in the paper. As a first approximation, we used the analytical fit of Koops et al's (2000) results derived in Ren and MacKenzie 2005 : Scr=2.349-T/259. This fit assumes that particles have a typical radius of 0.25 µm and that they freeze homogeneously within 1 min (see also Kärcher and Burkhardt 2008). In this equation, the solution droplets are assumed to be in equilibrium with the environment which is reasonable for

most atmospheric situations (Koop 2015) and particularly for temperatures > 205 K (which is the case for our two fog events at Dome C).

Values of Scr depend on the size of the particle, on the composition of the particle  and on the formulation  - and related uncertainties - of water activities and saturation vapor pressure (see thorough discussion in Baumgartner et al. (2022)). Individually, those effect make Scr vary by about 0.01 to 0.05 (see Baumgartner). An envelop of 0.05 has also been added around the Koop's curve in our Figs.4, 8 and B1. The inclusion of this shading is only intended as a rough indicator of the uncertainty and to guide the eye. Moreover, we have added the following paragraph in Section 2.2 :

*'To detect the possible occurrence of homogeneous freezing of solution aerosols, we will compare our RHi measurements with the so-called Koop et al. (2000)'s threshold. In the approach of Koop et al. (2000), solution particles spontaneously freeze when RHi exceeds a threshold value that primarily depends on temperature. As a first approximation, we calculate the RHi threshold value ($RHi_T$, in %) using the analytical fit of Koop et al. (2000)'s experimental results derived in Ren and Mackenzie (2005):*
*$RHi_T = (2.349 − T/259) \cdot 100$*

*where T is the temperature in Kelvin. This fit has been performed for solution particles in equilibrium with the ambient vapor that have a typical radius of 0.25 µm and that can freeze homogeneously within 1 min (see also Kärcher and Burkhardt, 2008). The exact value of the threshold also depends on the size of the particle as well as on the composition thereof and on theformulation and uncertainties of water activities and saturation vapor pressure. Individually, those effects make $RHi_T$ vary by about 1 to 5 % (see Baumgartner et al., 2022). An envelop of 5 % has therefore been added around the Koop's curve in our graphs. This envelop is only intended as a rough indicator of the uncertainty and to guide the eye.'*

Baumgartner, M., Rolf, C., Grooß, J.-U., Schneider, J., Schorr, T., Möhler, O., Spichtinger, P., and Krämer, M.: New investigations on homogeneous ice nucleation: the effects of water activity and water saturation formulations, Atmospheric Chemistry and Physics, 22, 65–91, https://doi.org/10.5194/acp-22-65-2022, 2022.

Kärcher B. and Burkhardt U. :A cirrus cloud scheme for general circulation models, Q. J. R. Meteorol. Soc. 134: 1439–1461 (2008)

Koop, T.: Atmospheric Water, in: Water: Fundamentals as the Basis for Understanding the Environment and Promoting Technology, edited by: Debenedetti, P. G., Ricci, A., and Bruni, F., IOS, Amsterdam, Bologna, 45–75, https://doi.org/10.3254/978-1-61499-507-4-45, 2015

Ren C, MacKenzie AR. 2005. Cirrus parameterization and the role of ice nuclei. Q. J. R. Meteorol. Soc. 131: 1585–1605.

I ask because the RHi of the measurements is close to but does not exactly meet the threshold. Or could it be that the freezing threshold is a bit lower at the warmer cirrus temperatures, as observed in the lab by  Schneider et al. (2021), ACP (https://acp.copernicus.org/articles/21/14403/2021/ ), and also discussed in Baumgartner et al. (2021)?

Note that for the second event, the threshold is met at z=18 m. For the 1st event, note that the difference between the homogeneous freezing threshold and the measured RHi peak is lower than the measurement uncertainty and that the Koop' threshold uncertainy as well (we now specify it in

the text).  Those uncertainties prevent us from discussing subtle variations in the freezing threshold or from questioning its exact value as in Schneider et al. (2021) and Baumgartner et al. (2022).

Also, I think it could be worth to discuss the long times where the
ice fog exist in slight subsaturation.... or could it be that the humidity has a slight dry bias ?

Following your recommmandation, for the first event we have adapted the text as follows.

For the first event :

*'Fig. 2a shows that the depth of the fog layer gradually increases from 0600 LT, 8 March up to about 80 m at 1800 LT, 8 March, as the daytime convective boundary layer deepens in $\sqrt{(t)}$ (Stull et al. 1990). The growth of the fog is possible in the higher part of the boundary layer as its top is supersaturated wrt ice (Fig 2c). Ice crystals can hence grow by vapour deposition and sediment down to the near-surface layers where they probably partly sublimate (Fig. 2 and 4). Concurring with Genthon et al. 2022 (see their Fig. 8), the near-surface air becomes subsaturated wrt ice particularly during daytime when the near-surface air warms by convective mixing.'*

For the second event :

*'From the evening of the 25 August, RHi remains slightly below saturation at 3 and 18 m probably owing to a net flux of vapor towards the surface in the very shallow boundary layer. Ice crystals at these two heights can therefore not grow very close to the surface but their detection at 18 m by the lidar may be rather explained by the sedimention from higher layers.'*